# Assessing Cross-Sectional Association of Uremic Pruritus with Serum Heavy Metal Levels: A Single-Center Study

**DOI:** 10.3390/diagnostics13233565

**Published:** 2023-11-29

**Authors:** Cheng-Hao Weng, Ching-Chih Hu, Tzung-Hai Yen, Ching-Wei Hsu, Wen-Hung Huang

**Affiliations:** 1Kidney Research Center, Department of Nephrology, Linkou Chang Gung Memorial Hospital, Taoyuan 333, Taiwan; drweng@seed.net.tw (C.-H.W.); m19570@cgmh.org.tw (T.-H.Y.); wei2838@gmail.com (C.-W.H.); 2Clinical Poison Center, Linkou Chang Gung Memorial Hospital, Taoyuan 333, Taiwan; 3College of Medicine, Chang Gung University, Taoyuan 333, Taiwan; nonahu5248@gmail.com; 4Graduate Institute of Clinical Medical Sciences, College of Medicine, Chang Gung University, Taoyuan 333, Taiwan; 5Department of Nephrology, Taoyuan Chang Gung Memorial Hospital, Taoyuan 333, Taiwan; 6Liver Research Unit, Department of Hepatogastroenterology, Chang Gung Memorial Hospital, Keelung 204, Taiwan

**Keywords:** heavy metals, pruritus, uremia, hemodialysis

## Abstract

(1) Background: Uremic pruritus (UP) is a common and taxing symptom in patients on maintenance hemodialysis (MHD). We have previously shown that blood lead levels (BLLs) and blood aluminum levels (BALs) were separately positively associated with UP in MHD patients. We also found that blood cadmium levels (BCLs) were positively associated with all-cause mortality and cardiovascular-related mortality in MHD patients. We wondered whether there is any correlation between BCLs and UP after adjusting for BLLs and BALs. (2) Methods: Patients enrolled in this study were all from three hemodialysis (HD) centers at Chang Gung Memorial Hospital, Lin-Kou Medical Center, including both the Taipei and Taoyuan branches. Correlations between UP and BLLs, BALs, BCLs, and other clinical data were analyzed. (3) Results: Eight hundred and fifty-three patients were recruited. Univariate logistic regressions showed that diabetes mellitus, hepatitis B virus infection, hepatitis C virus infection, HD duration, hemodiafiltration, dialysis clearance of urea, normalized protein catabolic rate, non-anuria, serum albumin levels, log (intact-parathyroid hormone levels), total serum cholesterol levels, serum low-density lipoprotein levels, log (blood aluminum levels), and log (blood lead levels) were associated with UP. Although log BCLs were not significantly associated with UP (*p* = 0.136) in univariate analysis, we still included log BCLs in multivariate logistic regression to verify their effect on UP given that our aim in this study was to verify associations between serum heavy metals and UP. Multivariate logistic regressions showed that log BLLs (OR: 27.556, 95% CI: 10.912–69.587, *p* < 0.001) and log BALs (OR: 5.485, 95% CI: 2.985–10.079, *p* < 0.001) were positively associated with UP. The other logistic regression, which stratified BLLs and BALs into high and low BLLs and BALs, respectively, showed that high BLLs or high BALs (low BLLs and low BALs as reference) (OR: 3.760, 95% CI: 2.554–5.535, *p* < 0.001) and high BLLs and high BALs combined (low BLLs and low BALs as reference) (OR: 10.838, 95% CI: 5.381–21.828, *p* < 0.001) were positively correlated with UP. (4) Conclusions: BLLs and BALs were positively correlated with UP. BCLs were not correlated with UP. Clinicians should pay more attention to the environmental sources of lead and aluminum to prevent UP.

## 1. Introduction

Uremic pruritus (UP) presents a prevalent and bothersome symptom among individuals undergoing maintenance hemodialysis (MHD). Its occurrence varies depending on factors such as country, dialysis method, specific dialysis facility, and the population being studied. UP impacts 26% to 48% of patients receiving hemodialysis [1]. There are several hypotheses and etiologies of UP, including systemic inflammation [2,3], imbalances in the expression of opioid receptors [4], and histamine and other pruritogens released by mast cells [5]. In our previous study, we showed that UP is an important predictor of 24-month cardiovascular mortality in MHD patients [6]. There are also significant associations between UP and poor quality of life and increased risk of mortality [1,7]. Therefore, recognition and treatment of UP in MHD patients is strongly indicated. In earlier research, we demonstrated a positive correlation between blood lead levels (BLLs) [8] and blood aluminum levels (BALs) [9] and UP in patients undergoing MHD, when considering these factors individually. We also found that the high blood cadmium level (BCL) group (>0.521 μg/L) had increased hazard ratios for all-cause mortality and cardiovascular-related mortality [10]. Therefore, we wondered whether there is any correlation between BCLs and UP after adjusting for BLLs and BALs. This study was designed to show whether the combinations of different BLLs, BALs, and BCLs can predict the occurrence of UP.

## 2. Material and Methods

### 2.1. Methods

The Institutional Review Board (IRB) Committee of Chang Gung Memorial Hospital approved the study protocol (Code of IRB: 98-1937B). The methods in this study were executed following the approved guidelines. Our IRB committee confirmed that informed consent was not necessary for this cross-sectional retrospective study. We gathered all primary data following the Strengthening the Reporting of Observational Studies in Epidemiology guidelines. Patient data were de-identified for privacy.

### 2.2. Patients

Data collection started in February 2016 and ended in November of the same year. All MHD patients in this study were recruited from three hemodialysis (HD) centers of Chang Gung Memorial Hospital in Linkou, Taoyuan, and Taipei. Maintenance hemodialysis patients who were 18 years old or older and had received HD for at least 6 months were enrolled in this study. Patients with malignancies, infectious diseases, and those who had been admitted or received surgery within 3 months were excluded from this study. Nearly all MHD patients received 4 h of HD per session and 3 HD sessions per week. Synthetic high-flux dialyzers were used for all MHD patients. Dialysate with standard ionic composition and a bicarbonate-based buffer was used in each HD session for all patients. The ultrapure water employed in dialysate preparation was generated through a combination of reverse osmosis and ultrafilters. Ultraviolet treatment was applied for the disinfection of the dialysis water. Cardiovascular diseases (CVDs) and smoking behavior were documented where applicable. UP was diagnosed according to the following criteria: pruritus emerging following HD with or without the use of anti-pruritic medications; evaluation carried out by skilled dermatologists or nephrologists. The extent of uremic pruritus was assessed using a visual analog scale (VAS), utilizing a 10 cm horizontal line where 0 points denoted no pruritus and 10 points represented the highest intensity of pruritus.

### 2.3. Measurement of BLLs and BCLs

BLLs and BCLs were assessed using a well-established approach involving electrothermal atomic absorption spectrometry (SpectrAA-200Z; Agilent Technologies, Santa Clara, CA, USA, www.agilent.com, accessed on 1 Feburary 2016). The measurements were conducted on the arterial side of the vascular access, just before the commencement of HD, following a 2 day HD-free interval [10,11,12].

### 2.4. Measurement of BALs

BALs were measured on the arterial side of the vascular access just before the start of HD following a 2 day HD-free interval. Quantification was carried out using the graphite furnace atomic absorption spectrometry method, utilizing a Perkin-Elmer 5100 atomic absorption spectrometer (Norwalk, CT, USA), as described in a previous study [9].

### 2.5. Definition of High and Low BLLs and BALs

We divided BLLs and BALs into high and low BLLs and BALs. High BLLs were defined as BLLs ≥ 12.77 µg/dL, according to our previous study [8]. High BALs were defined as BALs ≥ 2 µg/dL, according to our previous study [9].

### 2.6. Laboratory Parameters

Blood samples were collected from the arterial side of the vascular access just prior to the commencement of HD following a 2 day HD-free interval. Biochemical data were assessed using a standard laboratory protocol. The adequacy of HD was determined using the Daugirdas method and was expressed as Kt/V_urea_. In our HD centers, Kt/V_urea_ is calculated using the single pool method. Anuria was defined as a 24 h urine output of <100 mL, while non-anuria was defined as a 24 h urine output of ≥100 mL.

### 2.7. Statistical Analysis

The normality of distribution was assessed using the Kolmogorov–Smirnov test. Continuous variables were presented as mean ± standard deviation or median (interquartile range), while categorical variables were represented as numbers or percentages. Associations between categorical variables were examined using chi-square or Fisher’s exact test. For comparing two groups, the Mann–Whitney U test or Student’s *t*-test were utilized. Univariate and multivariate logistic regression analyses were employed to assess the variables associated with UP. Data analysis was conducted using SPSS, version 23 for Windows 10 (SPSS Inc., Chicago, IL, USA). The significance level was established at *p* < 0.05.

## 3. Results

The flow chart of MHD patient recruitment is demonstrated in Figure 1. Initially, 937 patients with MHD were chosen. However, 84 of these MHD patients did not have complete data and were subsequently excluded from this study. Eight hundred and fifty-three patients were recruited (Table 1). They had received an average HD duration of 6.98 ± 5.37 years. BLLs and BALs in patients with UP were significantly higher than in patients without UP (*p* < 0.001); however, serum ferritin levels and BCLs were not (Figure 2).

### 3.1. Predictors of Uremic Pruritus

Univariate logistic regressions with *p* value < 0.05 showed that non-diabetes mellitus (DM) [odds ratio (OR): 0.47, 95% confidence interval (CI): 0.30–0.75, *p* = 0.001], hepatitis B virus infection (HBV) (OR: 0.54, 95% CI: 0.29–0.98, *p* = 0.042), hepatitis C virus infection (HCV) (OR: 1.48, 95% CI: 1.01–2.19, *p* = 0.045), HD duration (OR: 1.11, 95% CI: 1.01–1.14, *p* < 0.001), hemodiafiltration (OR: 1.54, 95% CI: 1.06–2.24, *p* = 0.022), dialysis clearance of urea (Kt/V_urea_) (OR: 2.87, 95% CI: 1.76–4.69, *p* < 0.001), normalized protein catabolic rate (nPCR) (OR: 1.91, 95% CI: 1.05–3.49, *p* = 0.035), non-anuria (OR: 0.45, 95% CI: 0.28–0.72, *p* = 0.001), serum albumin levels (OR: 0.59, 95% CI: 0.37–0.94, *p* = 0.026), log (intact parathyroid hormone levels) (iPTH) (OR: 1.51, 95% CI: 1.13–2.02, *p* = 0.005), total serum cholesterol levels (OR: 1.01, 95% CI: 1.00–1.01, *p* = 0.017), low-density lipoprotein (LDL) (OR: 1.01, 95% CI: 1.00–1.01, *p* = 0.005), log BALs (OR: 5.6, 95% CI: 3.25–9.67, *p* < 0.001), and log BLLs (OR: 37.07, 95% CI: 15.35–89.56, *p* < 0.001) were associated with UP. To reduce omitted-variable bias, we used a higher *p* value cutoff, with *p* < 0.1. Univariate logistic regression identified 17 variables with *p* value < 0.1 that were associated with UP (Table 2). Although log BCLs were not significantly associated with UP (*p* = 0.136), we still included log BCLs in multivariate logistic regression to verify its effect on UP, given that our aim of this study was to verify associations between serum heavy metals and UP. Eighteen variables [age, DM, HBV, HCV, HD duration, hemodiafiltration, Kt/V_urea_, nPCR, non-anuria, serum albumin levels, corrected calcium levels, total serum cholesterol levels, log iPTH, LDL, log BALs, log BLLs, log BCLs, and log ferritin were introduced into multivariate logistic regression, which showed that the following were correlated with UP: HD duration (OR: 1.092, 95% CI: 1.055–1.131, *p* < 0.001), log BLLs (OR: 27.556, 95% CI: 10.912–69.587, *p* < 0.001), log ferritin (OR: 2.100, 95% CI: 1.387–3.180, *p* < 0.001), log BALs (OR: 5.485, 95% CI: 2.985–10.079, *p* < 0.001), and LDL (OR: 1.010, 95% CI: 1.004–1.016, *p* = 0.001) (Table 3).

### 3.2. Predictors of Uremic Pruritus Analyzed by High and Low BLLs and BALs

Multivariate logistic regression showed that HD duration (OR: 1.083, 95% CI: 1.045–1.122, *p* < 0.001), high BLLs or high BALs (low BLLs and low BALs as reference) (OR: 3.760, 95% CI: 2.554–5.535, *p* < 0.001), high BLLs and high BALs (low BLLs and low BALs as reference) (OR: 10.838, 95% CI: 5.381–21.828, *p* < 0.001), log ferritin (OR: 2.207, 95% CI: 1.459–3.340, *p* < 0.001), and LDL (OR: 1.011, 95% CI: 1.005–1.017, *p* = 0.001) were positively associated with UP. On the other hand, DM (OR: 0.581, 95% CI: 0.347–0.972, *p* = 0.039) and non-anuria (OR: 0.532, 95% CI: 0.310–0.912, *p* = 0.022) were negatively associated with UP (Table 4).

## 4. Discussion

We have previously showed that BLLs [8] and BALs [9] were separately positively associated with UP in MHD patients. The possible mechanisms of lead-induced pruritus are as follows: altering µ and δ receptors and biological responses to opioids [13]; induction of reactive oxygen species [14]; lipid peroxidation [15]; increase in β-2 microglobulin excretion [16]; and precipitation of calcium phosphate, found in various parts of the body, such as bones and skin [17]. The link between BALs and pruritus in MHD patients remains unclear. However, both our previous study and Friga’s study [18] demonstrated a positive correlation between UP and BALs in individuals undergoing MHD, with the intensity of pruritus significantly tied to BAL concentration. In the general population, sustained pruritus and the presence of itching nodules have been reported following the use of vaccines containing aluminum [19]. Bergfors et al. [20] highlighted a high incidence of pruritic nodules in 77% of children after receiving diphtheria–tetanus/acellular pertussis vaccines associated with aluminum allergy. Animal studies have also indicated that chronic exposure to low doses of lead results in the generation of reactive oxygen species (ROS) [21]. ROS have been recognized to play a role in atopic dermatitis, a non-contagious, recurring inflammatory skin condition characterized by eczema and pruritus [14]. Therefore, lead might induce UP through the initiation of ROS. Besides, MHD patients with both high BLLs and BALs had a significantly higher odds ratio, 10.838, for suffering from UP, as compared with MHD patients with low BLLs and BALs.

After starting HD, the high prevalence of elevated BALs, BCLs, and BLLs were noted [22]. According to our present study, MHD patients with both high BLLs and BALs were a strong predictor of UP. MHD patients need to reduce their exposure to heavy metals such as aluminum, cadmium, and lead. The sources of these heavy metals might be tobacco smoke, aluminum-based bakery products, aluminum-treated water, environmental sources of lead including petroleum, industrial production, paint, water pipes made of lead, food stored in cans, “moonshine” (homemade distilled ethanol), and food products containing a high level of heavy metals (rice, wheat, vegetables, and tea).

Aluminum has also been noted to induce oxidative stress, while lead and aluminum exposure also alters N-methyl-d-aspartate receptor (NMDAR) subunit expression in the brain [23]. NMDAR not only exits in the brain, but also in the skin. Haddadi et al. showed that intradermal injection of an NMDAR antagonist could modulate chloroquine (CQ), provoking scratching behavior in mice [24]. Therefore, lead and aluminum might also induce NMDAR in the skin of MHD patients and induce UP. A combination of high BLLs and high BALs would have synergistic effects on the induction of NMDAR in patients. Another etiology of heavy-metal-induced oxidative stress includes lead and aluminum damage to cellular components via elevated levels of oxidative stress and ∙OH free radicals through the Fenton reaction [25]. Therefore, both high BLLs and BALs might synergistically induce oxidative stress and ∙OH free radicals through the Fenton reaction.

In our previous study, serum ferritin levels was associated with UP [9]. Ferritin not only stores iron and releases it in a controlled fashion, but it will rise in the conditions of infection, inflammation and cancers [26]. Spada et al. showed that in MHD patients, ferritin levels could potentially contain metals other than iron, notably aluminum, considering the potential oral intake of aluminum by these patients [27]. Hence, there might be a positive correlation between BALs and blood ferritin levels in our MHD patients, as both were found to be linked with UP.

In this investigation, we demonstrated a positive association between serum LDL levels and UP. Elevated LDL levels are recognized to be linked with various dermatological conditions, most of which are chronic inflammatory disorders. The underlying mechanism often involves the secretion of proinflammatory cytokines. Research has highlighted an increased presence of LDL in skin conditions like psoriasis, lichen planus, pemphigus, granuloma annulare, histiocytosis, as well as connective tissue diseases like lupus erythematosus [28]. Many of these diseases manifest with pruritus as a symptom. Several mechanisms have been proposed to explain the connection between inflammation and heightened LDL levels, including the modulation of lipoprotein lipase (LPL) enzymatic activity through anti-LPL antibodies and reduced LPL activity due to various pro-inflammatory cytokines such as tumor necrosis factor-α, interleukin-1 (IL-1), IL-6, interferon-γ, and monocyte chemoattractant protein-1 [29].

DM was also a predictor of UP in this study. One of the etiologies of UP is neuropathy in MHD patients. Dialysis patients show an altered neurophysiological response. Further, peripheral cutaneous nerve endings rarify in uremia; however, they also sprout irregularly into the epidermis, possibly leading to easier excitability [30]. DM is an important cause of neuropathy in MHD patients [31] and might cause the above neurological changes.

Non-anuria was a negative predictor of UP in our study. Lengton et. al. found an inverse association between residual estimated glomerular filtration rate (eGFR) and pruritus at 12 months in both HD and peritoneal dialysis (PD) patients. After multivariable adjustment, each 1 mL/min/1.73 m^2^ higher residual eGFR was associated with a significantly lower risk of pruritus in HD patients [32]. Their study is compatible with our finding that anuria will positively predict UP.

Besides risk factors associated with UP discovered in this study, other risk factors associated with UP in MHD patients have been documented to include advanced age, gender, imbalances in calcium phosphate levels, extended dialysis duration, as well as comorbidities such as concurrent cardiovascular disease, congestive heart failure, pulmonary disease, liver disorders, neurological conditions, hepatitis C infection, hyperparathyroidism, inadequate dialysis, elevated serum magnesium, xerosis, and anemia [33].

This study had several limitations that warrant discussion. First, it is essential to note that the observed associations of BLLs and BALs with UP do not necessarily imply a causal relationship. To investigate a causal connection, a clinical trial employing chelation therapy, such as calcium disodium ethylenediaminetetraacetic acid, should be conducted on patients with MHD who exhibit elevated BLLs or BALs. This trial would help determine whether UP improves after reducing BLLs or BALs through chelation therapy. Secondly, our data collection for clinical parameters and heavy metal levels was cross-sectional in nature. To better understand the dynamics of UP severity in relation to changes in heavy metal levels, it is imperative to gather time-series data, which would allow us to assess how UP evolves over time as heavy metal levels fluctuate. Third, the distribution of heavy metal levels in our study deviated from a normal distribution. Expanding the study by recruiting a larger cohort of MHD patients could help determine whether heavy metal levels tend to approach a more normal distribution in a broader sample.

## 5. Conclusions

In conclusion, BLLs and BALs were positively associated with UP. BCLs were not correlated with UP. Patients on MHD should pay more attention to the environmental sources of lead and aluminum to prevent UP.

## Figures and Tables

**Figure 1 diagnostics-13-03565-f001:**
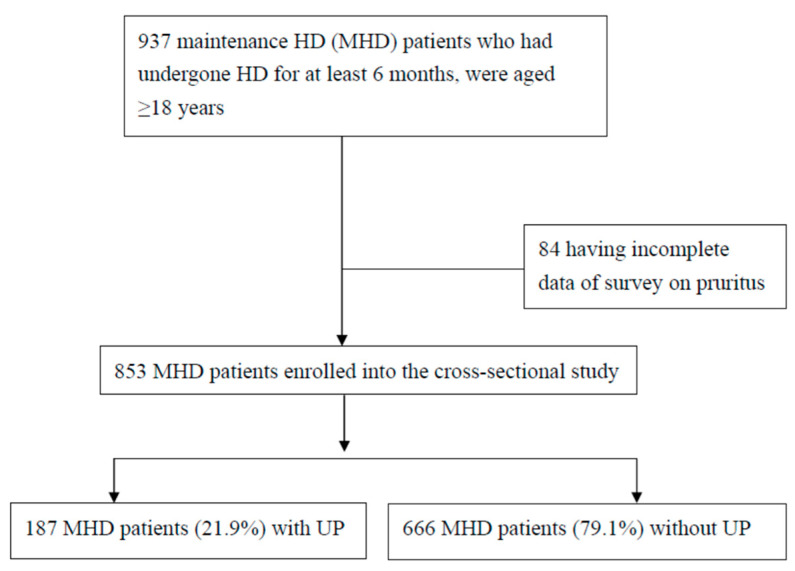
Flow chart of patient recruitment.

**Figure 2 diagnostics-13-03565-f002:**
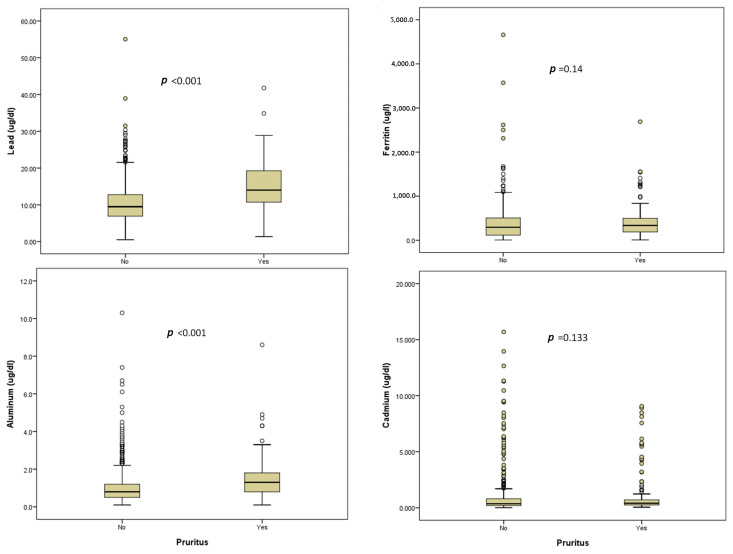
Box whisker plots show the serum lead, ferritin, aluminum, and cadmium levels of patients in the absence of UP (*n* = 666) and in its presence (*n* = 187); the data were compared. Whiskers on the box plot represent the 95% confidence intervals, while dots indicate outliers. The lower and upper edges of the boxes correspond to the 25th and 75th percentiles, respectively. The solid line within each box represents the median. Significantly elevated levels of serum lead and aluminum were observed in patients with UP compared to those without UP (*p* < 0.001); however, serum ferritin and cadmium were not. Data were compared using the Mann–Whitney U test.

**Table 1 diagnostics-13-03565-t001:** Characteristics of studied MHD patients.

Characteristics	Total (853) Mean ± SD/Median (Range)
Demographics	
Age (years)	56.17 ± 13.59
Male sex (yes)	434 (50.9%)
Body mass index (kg/m^2^)	22.18 ± 3.19
Smoking (yes)	146 (17.1%)
Co-morbidity	
Diabetes mellitus (yes)	188 (22%)
Hypertension (yes)	335 (39.3%)
Previous CVD (yes)	40 (4.7%)
HBV (yes)	95 (11.1%)
HCV (yes)	166 (19.5%)
Uremic pruritus (yes)	187 (21.9%)
Dialysis-related data	
Hemodialysis duration (years)	6.98 ± 5.37
Erythropoietin (U/kg/week)	73.54 ± 47.24
Fistula as blood access (yes)	679 (79.6%)
Hemodiafiltration (yes)	185 (21.7%)
Kt/V_urea_ Daugirdes	1.8 ± 0.32
nPCR (g/kg/day)	1.19 ± 0.27
Residual daily urine of >100 mL	175 (20.5%)
Biochemical data	
Hemoglobin (g/dL)	10.52 ± 1.36
Albumin (g/dL)	4.07 ± 0.35
Creatinine (mg/dL)	10.89 ± 2.4
Corrected calcium (mg/dL)	9.94 ± 0.93
Phosphate (mg/dL)	4.84 ± 1.36
Intact parathyroid hormone (pg/mL) *	129.3 (52.45, 311.1)
hsCRP (mg/L) *	2.92 (1.4, 7.0)
Cardiovascular risks	
Cholesterol (mg/dL)	171.22 ± 37.22
Triglyceride (mg/dL)	164.6 ± 115.34
LDL (mg/dL)	94.74 ± 30.57
Heavy metals	
Ferritin (μg/L) *	305.7 (129.3, 505.4)
BLLs (μg/dL) *	10.47 (7.26, 14.27)
BCLs (μg/dL) *	0.37 (0.21, 0.81)
BALs (μg/dL) *	0.9 (0.6, 1.4)

Abbreviations: CVD: cardiovascular disease; HBV: hepatitis B virus infection; HCV: hepatitis C virus infection; nPCR: normalized protein catabolic rate; hsCRP: high-sensitivity C-reactive protein; LDL: low-density lipoprotein; Kt/V_urea_: dialysis clearance of urea; BLLs: blood lead levels; BCLs: blood cadmium levels; BALs: blood aluminum levels; * non-normal distribution data are presented as median (interquartile range).

**Table 2 diagnostics-13-03565-t002:** Univariate logistic regression analysis between uremic pruritus and clinical variables.

Characteristics	Univariate Logistic Regression	
Variables	Odds Ratio (OR) 95% Confidence Intervals (CI)	*p*
Age (years)	1.01 (1.0–1.02)	0.059
Male sex	0.82 (0.59–1.14)	0.237
Body mass index (kg/m^2^)	0.98 (0.93–1.03)	0.513
Smoking (yes)	0.91 (0.58–1.4)	0.659
Non-diabetes mellitus (yes)	0.47 (0.30–0.75)	0.001
Hypertension (yes)	0.99 (0.71–1.38)	0.94
Previous CVD (yes)	0.75 (0.33–1.71)	0.49
HBV (yes)	0.54 (0.29–0.98)	0.042
HCV (yes)	1.48 (1.01–2.19)	0.045
Hemodialysis duration (years)	1.11 (1.07–1.14)	<0.001
Fistula as blood access (yes)	1.38 (0.89–2.11)	0.143
Hemodiafiltration (yes)	1.54 (1.06–2.24)	0.022
Kt/V_urea_ (Daugirdes)	2.87 (1.76–4.69)	<0.001
nPCR (g/kg/day)	1.91 (1.05–3.49)	0.035
Non-anuria	0.45 (0.28–0.72)	0.001
Hemoglobin (g/dL)	1.07 (0.95–1.2)	0.245
Serum albumin (g/dL)	0.59 (0.37–0.94)	0.026
Creatinine (mg/dL)	0.99 (0.93–1.06)	0.835
Corrected calcium (mg/dL)	1.18 (0.99–1.41)	0.058
Phosphate (mg/dL)	1.01 (0.89–1.13)	0.93
Log iPTH	1.51 (1.13–2.02)	0.005
Log hsCRP	0.93 (0.67–1.28)	0.644
Cholesterol (mg/dL)	1.01 (1.00–1.01)	0.017
Triglyceride (mg/dL)	0.99 (0.98–1.00)	0.184
LDL (mg/dL)	1.01 (1.00–1.01)	0.005
Log BALs	5.6 (3.25–9.67)	<0.001
Log BLLs	37.07 (15.35–89.56)	<0.001
Log BCLs	1.30 (0.92–1.84)	0.136
Log ferritin	1.37 (0.96–1.95)	0.079

Abbreviations: HBV: hepatitis B virus infection; HCV: hepatitis C virus infection; nPCR: normalized protein catabolic rate; iPTH: intact parathyroid hormone; hsCRP = high-sensitivity C-reactive protein; LDL = low-density lipoprotein; Kt/V_urea_ = dialysis clearance of urea; BLLs = blood lead levels; BCLs = blood cadmium levels; BALs = blood aluminum levels.

**Table 3 diagnostics-13-03565-t003:** Multivariate logistic regression analysis (forward method) between uremic pruritus and heavy metals and other variables.

	Multivariate Logistic Regression	
Variables	Odds Ratio (OR), 95% Confidence Intervals (CI)	*p*
Hemodialysis duration (years)	1.09 (1.06, 1.13)	<0.001
Log BLLs	27.56 (10.91, 69.59)	<0.001
Log ferritin	2.10 (1.39, 3.18)	<0.001
Log BALs	5.49 (2.99, 10.08)	<0.001
LDL (mg/dL)	1.01 (1.00, 1.02)	0.001

After adjustment for age, DM, HBV, HCV, hemodiafiltration, Kt/V_urea_, serum albumin levels, corrected calcium, nPCR, non-anuria, log iPTH, and log BCLs. Abbreviations: DM: diabetes mellitus; HBV: hepatitis B virus infection; HCV: hepatitis C virus infection; nPCR, normalized protein catabolic rate; iPTH: intact parathyroid hormone; LDL: low-density lipoprotein; Kt/V_urea_: dialysis clearance of urea; BLLs: blood lead levels; BALs: blood aluminum levels; BCLs: blood cadmium levels.

**Table 4 diagnostics-13-03565-t004:** Multivariate logistic regression analysis (forward method) between uremic pruritus and high BLLs and/or high BALs and other variables.

	Multivariate Logistic Regression	
Variables	Odds Ratio (OR) 95% Confidence Intervals (CI)	*p*
Hemodialysis duration (years)	1.08 (1.05, 1.12)	0.000
Non-DM	0.5 (0.35, 0.97)	0.039
Non-anuria	0.53 (0.31, 0.91)	0.022
BLLs and BALs		
Low BLLs and low BALs (reference)		
High BLLs or high BALs	3.76 (2.55, 5.54)	0.000
High BLLs and high BALs	10.84 (5.38, 21.83)	0.000
LDL (mg/dL)	1.01 (1.01, 1.02)	0.001
Log ferritin	2.21 (1.46, 3.34)	0.000

After adjustment for age, HBV, HCV, hemodiafiltration, Kt/V_urea_, serum albumin levels, corrected calcium, nPCR, log iPTH, and log BCLs; high BALs: blood aluminum levels ≥ 2 µg/dL; high BLLs: blood lead levels ≥12.77 µg/dL; low BLLs and low BALs: 501 subjects; high BLLs or high BALs: 306 subjects; high Pb and high Al: 46 subjects. Abbreviations: DM: diabetes mellitus; HBV: hepatitis B virus infection; HCV: hepatitis C virus infection; nPCR, normalized protein catabolic rate; iPTH: intact parathyroid hormone; LDL: low-density lipoprotein; Kt/V_urea_: dialysis clearance of urea; BCLs: blood cadmium levels.

## Data Availability

Data is contained within the article.

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
