# Peer review of "Assessing Cross-Sectional Association of Uremic Pruritus with Serum Heavy Metal Levels: A Single-Center Study"

_diagnostics, 2023, doi:10.3390/diagnostics13233565_

Round 1

Reviewer 1 Report

Comments and Suggestions for Authors

The authors examine associations with heavy metal serum levels and uremic pruritus in a large cohort of hemodialysis patients. Overall, I struggle to find any real new information in the manuscript given the two previous papers by the same authors (Hsu et al. Sci Rep. Association Between Serum Aluminum Level and Uremic Pruritus in Hemodialysis Patients 2018 Nov 22;8(1):17251.  doi: 10.1038/s41598-018-35217-6 and Blood lead level is a positive predictor of uremic pruritus in patients undergoing hemodialysis Ther Clin Risk Manag. 2017 Jun 12:13:717-723.  doi: 10.2147/TCRM.S135470.  eCollection 2017.). There is very little focus on cadmium and most of the manuscript deals with the association between aluminum/lead and uremic pruritus which has already been reported in the two previous papers. I find it questionable why the authors do not focus more on cadmium rather than repeating previous reported associations with aluminum and lead. I also have some statistical questions which needs clarification as outlined in the following.

Abstract:

The abstract should contain information on the number of patients in the study (n=853).

Introduction:

The use of abbreviations is not consistent and inconsequent.  E.g., the authors do not use the abbreviation UP introduced for uremic pruritus. When they state Uremic pruritus impacts 26% to 48% of patients…

I also suggest that the authors introduce the abbreviation HD for hemodialysis in the introduction.

Rephrase/rewrite the sentence “We have previous showed…”

Methods

The definition of uremic pruritus is quite vague. A VAS-scale was used for grading pruritus intensity (0-10) yet this intensity is not used at all why not? Was there any correlation between pruritus severity and the blood levels of heavy metals (lead, aluminum and cadmium)?

Statistical analysis

There is no description of why the heavy metals (aluminum, lead, and cadmium) and other variables were log transformed. I suspect it was due to a non-normal distribution. Likewise, it is not clear when the authors used Mann-Whitney U test or Student’s t-test. The latter requires a normal distribution. Use of Pearson correlation coefficients and interpretation of the odd ratios is also somewhat dubious as outlined in the comments below.

Result section

Why is pruritus severity not used at all? Was there any correlation between pruritus severity and the blood levels of heavy metals or other significant predictors?

Table 1

I would like demographics according to the presence (n=187) or absence (n=666) of uremic pruritus. Do the patients with uremic pruritus differ markedly in terms of comorbidity and dialysis related parameters?  

3.1

Correlation between BAL, BLL, and BCL with UP.

I find it somewhat strange that the authors use the Pearson Correlation coefficient as a measure for a dichotomized outcome such as uremic pruritus (UP). Usually, the Pearson correlation coefficient is used to measure the linear correlation between two set of data with a value between -1 and 1. I would advocate the use of univariate logistic regression with UP as outcome (binary dependent variable) and BAL, BLL, and BCL as continuous variables (predictors) like the authors present in Table 2.

I suggest the authors remove this section and write more about Figure 1. 

Figure 1 

Is not mentioned in the text but essentially shows what the authors state in section 3.1. Why not?

Regarding cadmium:

Why do the authors not use a dichotomized approach/or cadmium above a certain threshold when interpreting cadmium levels and the association with uremic pruritus like the approach used for lead and aluminum?

Regarding diabetes as predictor for UP:

In Table 2 with univariate logistic regression Diabetes mellitus (yes) has OR 0.47(0.30-0.75) and non-anuria has OR 0.45(0.28-0.72).  In Table 4 with multivariate logistic regression the authors found that diabetes and non-anuria had OR 0.58(0.34-0.97) and OR 0.53(0.31-0.91) for UP. That implies that patients with diabetes and no urine production is less likely to have UP. I suspect that the authors have reported it wrong so that it should be non-diabetes patients that has OR (univariate) 0.47 (0.30-0.75) OR (multivariate) 0.58(0.34-0.97).

In the discussion non-anuria is described as a negative predictor of UP. Yet, if the OR for diabetes is correct the same should be the case for diabetes. This must be clarified.

Do the authors think HBV is protective of uremic pruritus given the OR of 0.54(0.29-0.98)?

Somewhat strange that a similar condition such as HCV has OR 1.48(1.01-2.19). Is this a change finding or similar to diabetes is it absence of HBV that should have OR 0.54?

Table 3 and 4

Why do the authors use 3 digits after the comma? E.g., (Table 3/4) Hemodialysis duration (years) 1.092 (1.055, 1.131) /1.083 (1.045, 1.122) why not simply 1.09 (1.06-1.13)/ 1.08(1.05-1.12) like Table 2? I suggest use of only two digits after the comma.

Table 4 introduces a new abbreviation Pb for blood lead not used elsewhere that is confusing. Why not keep it as BLL rather than Pb?

Section 3.3.

Definition of high BLL and high BAL belongs in the methods section. Use of BLL/BAL as a binary predictor (low vs. high) rather than BLL/BAL as a continuous variable is ok (also High BLL or high BAL and High BLL and high BAL, respectively). Also, I suspect that the references given for the definition of high BLL and BAL is incorrect. Please check if they are correct. 

Discussion section:

Sentence starting with “After starting hemodialysis, the high prevalence of…” Is there a word missing? Meaning is not entirely clear. Elevated levels?

Sentence “Aluminum was also been noted…” Is there a word missing/rephase meaning is not clear.

Perhaps change to “Aluminum has also been noted to…”

Sentence starting with “N-methyl-d-aspartate receptor could be simplified to NMDAR.

The abbreviation HD for Hemodialysis is not consistently used. HD as abbreviation should be introduced earlier in the introduction.

The use of the abbreviation MHD for maintenance hemodialysis is also not consistently used and it is repeated in the discussion, which is not necessary se also previous comment.

 Regarding ferritin levels. Do the authors believe that other heavy metals are built into the ferritin molecule or is high ferritin as an acute phase reactant reflecting elevated levels of heavy metals? Why do the authors not test in their cohort whether high ferritin is associated with high BLL or BAL levels? Thus, were ferritin levels significantly higher among patients with high BLL and/or BAL?

 Regarding diabetes as a predictor for UP. Please see previous comment regarding the interpretation of the OR for diabetes.

There is little reflection regarding limitations of the study. Associations between heavy metals and pruritus does not necessarily imply causality, yet this is not discussed at all.

Are Taiwanese hemodialysis patients comparable to patients on hemodialysis treatment in Europe and the US (exposure to heavy metals, heavy metals content in food and water etc.)?

Comments on the Quality of English Language

Should be improved. Several words missing or meaning obscured. Inconsistent use of abbreviations is also a problem.

Reviewer 2 Report

Comments and Suggestions for Authors

1.The determination of the number of samples should be described in detail.

2.Create a technology roadmap.

3.The diagnosis of uremic pruritus based on the emergence of itching after dialysis can be subjective and dependent on the patient's report and the professional's judgment. Additionally, the lack of specific criteria for the use of anti-pruritic medications may introduce variability in the patient's experience of itching.

4.Although the assessment by dermatologists or nephrologists is mentioned, there is no detailed explanation of their training or standardization in the use of the Visual Analog Scale (VAS) for this specific symptom, which could result in variability between observers.

5.Potential Selection Bias: The study population comes from a specific hospital system and may not represent the broader group of patients undergoing maintenance hemodialysis (MHD).

Reviewer 3 Report

Comments and Suggestions for Authors

The paper “Assessing Cross-Sectional Association of Uremic Pruritus with Heavy Metal Serum Levels: A Single-Center Study” submitted to revision consists of interesting research about the possible role of heavy metals on the development of Uremic Pruritus in patients on Dialysis treatment. The paper is quite original and might bring some new knowledge in the pathogenesis of UP, but there are some questions to which the authors have to give some explanations. 

1.       In the paper, there are no explanations about the technological supplies treating tap water for its use in dialysate preparation

2.       The authors have ever tested the heavy metal concentration in water for dialysate?

3.       It is well known that the concentrated solution for dialysate preparation can be contaminated with metals secondary to the technological processes of dialysate preparation

4.       What hypotheses do the authors make about the source of heavy metal contamination?

5.       Regarding aluminum were there any patients taking phosphorus chelators containing aluminum?

              6 .    There was any correlation of UP with HCV positive test or hepatic disease?

Round 2

Reviewer 2 Report

Comments and Suggestions for Authors

It is suggested by the results that environmental metal pollution should be further surveyed, as the higher the aluminum concentration the tap water, the lower the metal concentration.